# Role of Yeasts on the Sensory Component of Wines

**DOI:** 10.3390/foods11131921

**Published:** 2022-06-28

**Authors:** Patrizia Romano, Giacomo Braschi, Gabriella Siesto, Francesca Patrignani, Rosalba Lanciotti

**Affiliations:** 1Faculty of Economy, Universitas Mercatorum, 00186 Rome, Italy; pot2930@gmail.com (P.R.); gasiesto@gmail.com (G.S.); 2Department of Agricultural and Food Sciences, Alma Mater Studiorum, University of Bologna, Campus Food Science, p.zza Goidanich 60, 47521 Cesena, Italy; giacomo.braschi2@unibo.it (G.B.); rosalba.lanciotti@unibo.it (R.L.); 3Interdepartmental Centre for Agri-Food Industrial Research, University of Bologna, Quinto Bucci 336, 47521 Bologna, Italy

**Keywords:** wine aroma, *Saccharomyces cerevisiae* biodiversity, yeast interaction, non-*Saccharomyces* yeasts, non-thermal technologies

## Abstract

The aromatic complexity of a wine is mainly influenced by the interaction between grapes and fermentation agents. This interaction is very complex and affected by numerous factors, such as cultivars, degree of grape ripeness, climate, mashing techniques, must chemical–physical characteristics, yeasts used in the fermentation process and their interactions with the grape endogenous microbiota, process parameters (including new non-thermal technologies), malolactic fermentation (when desired), and phenomena occurring during aging. However, the role of yeasts in the formation of aroma compounds has been universally recognized. In fact, yeasts (as starters or naturally occurring microbiota) can contribute both with the formation of compounds deriving from the primary metabolism, with the synthesis of specific metabolites, and with the modification of molecules present in the must. Among secondary metabolites, key roles are recognized for esters, higher alcohols, volatile phenols, sulfur molecules, and carbonyl compounds. Moreover, some specific enzymatic activities of yeasts, linked above all to non-*Saccharomyces* species, can contribute to increasing the sensory profile of the wine thanks to the release of volatile terpenes or other molecules. Therefore, this review will highlight the main aroma compounds produced by *Saccharomyces* *cerevisiae* and other yeasts of oenological interest in relation to process conditions, new non-thermal technologies, and microbial interactions.

## 1. Introduction

Wine aroma is a very complex concept since it is provided by a hundred different compounds with concentrations varying between 10^−1^ and 10^−10^ g/kg. Of course, the balance and interaction of all the present compounds (volatile or not) determine the wine aromatic quality [1,2,3]. However, the most common classification of wine aromas divides them into three main classes according to their origins: varietal compounds from grapes (primary aromas), those resulting from the fermentation metabolism of wine microorganisms (secondary aromas), and those from aging in oak barrels and bottles (tertiary aromas) (Figure 1).

Among all the compounds present in wine, volatile organic compounds (VOCs), with different polarities and volatilities, produced in widespread concentrations ranging from ng to hundreds of mg/L [4,5], have an aromatic value and play a central role in defining wine sensorial identity. They are, in fact, perceived in the nose and, when ingested, through the retro-sense of smell, they provide sensations that can be positive or negative, determining wine valorization [6].

Primary or varietal aromas are defined as compounds that derive directly from the variety of grapes and, for this reason, directly contribute to wine typicity. While the qualitative varietal aroma depends on the grapevine cultivar, the amounts present in the grape are strictly related to the environmental and pedo-climatic conditions. The compounds involved in the varietal aroma of the cultivars of *Vitis vinifera* are essentially: terpenes, norisoprenoids, and methoxypyrazine [7,8]. Secondary aromas are the compounds produced by yeasts during the fermentation process. Due to their sensory characteristics, secondary aromas are responsible for the vinous and fruity (in some cases) notes of wines. The compounds that directly contribute to the wine secondary aromas are higher alcohols, acetic and ethyl esters, and, finally, carbonyl and sulfur compounds. The formation of secondary aromas, as well as their accumulation in relation to the microbial consortium used for winemaking, will be deeply discussed in the subsequent paragraph. Tertiary aromas are related to wine aging, and the process generally consists of two phases: maturation (oxidative aging) and bottle aging (reductive aging). Oxidative aging commonly takes place in an oak barrel, and, for this reason, the volatile compounds extracted from oak woods play an important role in the wine aromas definition. Notable among them are: oak lactones (E)/(Z)-β-methyl-γ-octalactone (associated with coconut notes) and vanillin (oaky and vanilla aromas). Bottle aging (or reduction ageing) is mainly associated with reductive thiols accumulation (2-furanmethanethiol), as well as sulfides (dimethyl sulfide) [9].

The secondary aroma, called fermentation aroma, represents the main part of the transformation of grape must into wine and is operated by yeasts. The fermentation of grape must into wine involves the conversion of grape sugars to ethanol and carbon dioxide through the metabolism of yeasts, which, during alcoholic fermentation, produces other compounds besides ethyl alcohol and carbon dioxide. In particular, the so-called “secondary aromas” are linked to the activity of yeasts as they can only occur during the transformation of grape must into wine [10,11]. Fermentation affects the release and development of many compounds, so it is reasonable to assume that both inoculated yeasts and indigenous yeasts found on skins and in cellar environments have a significant effect on wine aroma. The type and quantity of the volatile substances synthesized depend on many factors, such as the temperature conditions and nitrogen concentration in the grape must; however, the influence of yeasts is one of the main agents affecting the final wine aroma profile [10]. Therefore, a selection criterion for a winning choice of a starter culture is based on the production levels of the different metabolites that form the “fermentation bouquet” and determine the complex sensorial and organoleptic character of the wine produced [12].

Natural grape must is hostile to microbial growth due to its low pH and high sugar concentrations. Furthermore, high concentrations of the antimicrobial sulfur dioxide are also added in most industrial fermentations, thus increasing the difficult conditions. Consequently, as fermentation proceeds, stress multiplies for additional reasons, including anaerobic conditions, depletion of nutrient reserves (nitrogen, lipids, and vitamins), increased acid concentrations, ethanol toxicity, and temperature changes. As *Saccharomyces cerevisiae* is able to overcome all fermentation stresses, it, despite being in very low populations in vineyards or grapes, becomes prevalent in fermentation and dominates over the other yeast species abundant in natural must [13,14]. This is the reason why *S. cerevisiae* has been designated with the title of “wine yeast”, being the main workhorse of the world wine industry [14].

As far as aroma is concerned, higher alcohols, esters, aldehydes, and terpenes are among the compounds that have the greatest impact on the “fermentation bouquet” [15]. The main aroma compounds produced during fermentation include alcohols, esters, acids, and carbonyl compounds [16].

### 1.1. Compounds from Primary Metabolism

Although the secondary metabolism is of fundamental importance in the generation of the sensory profile of a wine, the aromatic bouquet of a wine is the result of a complex interaction between numerous volatile chemical compounds and the macromolecules of the system. From a quantitative point of view, the metabolites that originate as products or by-products of the glycolytic process and alcoholic fermentation are undoubtedly the most abundant and mainly include ethanol, glycerol, acetic acid, and succinic acid [17].

Although they have different, but still very high, odorosity thresholds, due to their presence at high levels, they are able to directly or indirectly affect the final aroma of the wine, influencing the effect of secondary metabolites on the sensory profile of the wine. This is the case for succinic acid, a compound absent in the raw material and produced by yeast during fermentation, which is able to modify the interactions between macromolecules of the system (for example, mannoproteins) and aromatic compounds modifying the vapor pressure and, consequently, also the perception of the latter. Some studies have shown, for example, that, in a wine model system, the reduction in the concentration of ethanol from 10 to 7% determined a marked increase in the odorous perception of fruity and floral [18]. Another study has shown how, by reducing the alcohol content of wine, its aromatic bouquet is influenced by the new balances that are established between molecules of different natures. On the other hand, however, following the sudden climatic changes, which affect the quality of the grapes and wine, it has been seen that one of the most important effects on the generation of the final bouquet is linked to the continuous increase in sugar content in the grapes, which translates into osmotic stress for the yeast and in higher concentrations of ethanol but also for acetic acid and glycerol.

High levels in ethyl alcohol determine a different perception of the final bouquet since they increase the herbaceous hints compared to the fruity ones, also causing a greater astringency on the part of tannins and, therefore, bitter notes. In addition, the high presence of ethyl alcohol can activate a stress response in the yeast responsible for fermentation, influencing both the expression of specific genes and the modulation of the composition and fluidity of the cell membrane. However, market trends, inevitably linked to consumer preferences, are those preferring products with reduced alcohol content [18]. This purpose in the oenological field could be pursued through a careful selection of the starter strains of *S. cerevisiae* or, preferably, with the strategy of co-inoculations or scalar fermentations with non-*Saccharomyces* yeasts that will be subsequently described [19]. The amount of compounds that can contribute to the aromatic impact of a wine represents about 1% of the components present in the wine as a whole. In any case, these are molecules with very low thresholds of perception that, even if present in minimal quantities, can significantly impact the generation of the sensory profile of the final product [20].

### 1.2. Compounds from Secondary Metabolism

As previously described, some of these compounds originate from the primary metabolism of yeasts; others are instead generated by the regulation of the secondary metabolism of the latter and follow peculiar metabolic pathways, as reported in Figure 2. In the secondary metabolism of yeasts, different classes of compounds, including higher alcohols, volatile fatty acids and esters, carbonyl compounds, sulfur compounds, terpenoids, and volatile phenols, can originate.

#### 1.2.1. Esters and Alcohols

Associated with floral scents, esters are the group of secondary aroma compounds with the highest importance in the wine. Formed by the combination of alcohols and organic acids with the elimination of water, esters are produced both during fermentation and aging processes. During alcoholic fermentation, esters produced by enzymes are divided into two main categories: acetate esters and ethyl esters [15,18]. Acetate esters, responsible of fruity aromas of wines, include an acid group derived from acetic acid while the alcoholic function is formed by ethanol, or a higher alcohol derived from amino acid metabolism. Conversely, ethyl esters are the result of a dehydration reaction between a small-, medium- (C6–C10), or branched-chain fatty acid and ethanol [15].

Regarding higher alcohols (or fusel alcohols), together with acetic and ethyl esters, they are the most important metabolites for defining wine aromas. The effect of fusel alcohols on wine aroma is dose-dependent and, for these reasons, an excess could negatively affect the wine aroma profiles.

Their presence and concentration in wine is influenced by various factors, including grapevine cultivation methods, as well as the yeast strain used in fermentation. Other parameters, such as degree of grape ripeness, grape must pH, the presence of nitrogenous substances (including amino acids), the quantity of suspended solids, must aeration conditions, and the fermentation temperature, can also play a fundamental role for the qualitative and quantitative accumulation of higher alcohols in the final products [15]. The biosynthesis of fusel alcohols by yeasts cells takes place through the metabolic pathway known as the Ehrlich pathway and involves amino acids. Higher alcohol biosynthesis starts with a transamination reaction, which involves an amino acid and α-ketoglutaric acid as the amino group acceptor. The reaction leads to the formation of glutamate (from α-ketoglutaric acid) and produces α-ketoacid from the substrate. The latter then undergoes decarboxylation (by a pyruvate decarboxylase), with the formation of an aldehyde, which, through an NADH-dependent alcohol dehydrogenase, forms the corresponding higher alcohol or is oxidized to the corresponding fusel acid [15].

The fusel acid formations occur predominantly in aerobic condition (fusel aldehyde is oxidated in a NAD+-dependent reaction), while fusel alcohols formation is favored by anaerobic conditions [21,22]. Since the Ehrlich pathway is far from simple, the production of higher alcohols involves a vast number of genes and Ehrlich pathway proteins. In the Ehrlich pathway, the transamination reaction is the crucial step for higher alcohols formation: depending on the amino acid involved in the reaction, a different product will be obtained. Branched-chain higher alcohols, such as isoamyl alcohol, activated amyl alcohol, and isobutanol, are obtained from leucine, isoleucine, and valine, respectively. Moreover, aromatic amino acids, such as tyramine and phenylalanine, can also generate tyrosol and phenylethyl alcohol, respectively, through the Ehrlich pathway. Although the formation of higher alcohols by the Ehrlich pathway is linked with amino acids availability, there is often no direct link between higher alcohol production and its amino acid precursor present in the grape must since the process is affected by the intracellular amino acids pool [15].

This could be explained by the physiological function of the Ehrlich pathway. Apparently, this metabolic pathway appears as a catabolic process for the degradation of amino acids, but the decarboxylation and reduction that lead to the formation of higher alcohols are related to the amino acid biosynthetic processes. One of the most accredited hypotheses is that Ehrlich’s pathway is a part of a scavenger system for the intracellular detoxification of excess of α-ketoacid resulting from sugars metabolism, especially in the fermentation process of musts with a high sugar concentration, as well as a metabolism/catabolism regulation system of amino acids [15]. In this perspective, it is primarily responsible for the formation of higher alcohols, esters, and aldehydes [23].

The population of *S. cerevisiae* present in spontaneous fermentation is characterized by a high genetic polymorphism that is made up of genetically different strains and, therefore, potentially capable of influencing the organoleptic and sensorial characteristics of the finished product. Therefore, it should be emphasized that different strains of *S. cerevisiae* exhibit a high degree of variability in all technological characteristics [24,25] and each fermentation is characterized by the presence and activity of different strains of this species [26,27,28]. The complex genetic patrimony of the species *S. cerevisiae* accompanies different wild strains with the same characteristics (same genes that code for the various characters) but at a considerably different level and, as such, they produce, from the point of view of flavor and taste, wines with very different characteristics, even though they come from the transformation of the same grape must [29]. In *S. cerevisiae*, the production of higher alcohols involves a vast number of genes and Ehrlich pathway proteins. The initial transamination step involves at least four genes. *Twt1p* (also known as *Bat1p* or *Eca39p*) and *Twt2p* (*Bat2p* or *Eca40p*) encode, respectively, for two branched-chain amino acid aminotransferases. The first (*Twt1p*) has a mitochondrial localization, while the second (*Twt2p*) is a cytosolic isozyme [22]. Batch *S. cerevisiae* cultures express the mitochondrial enzyme during the exponential growth phase, while it is repressed during the stationary growth phase. By contrast, the *Twt2p* cytosolic form has the opposite regulatory pattern [22]. Collection of *S. cerevisiae* wine strains from spontaneous fermentations has demonstrated the existence of such strong polymorphism within this species [30]. Therefore, the production level of by-products is considered as an individual strain characteristic, underlining the importance of characterizing strains for industrial purposes [31].

From a quantitative point of view, higher alcohols are the most important group of compounds that *S. cerevisiae* produces during fermentation through the Ehrlich pathway [32]. Typical representatives of higher alcohols in wine are 1-propanol, 1-butanol, isobutanol, 2-phenylethanol, and isoamyl alcohol [33]. Table 1 reports the odor corresponding to the different higher alcohols and esters. If the total concentration of higher alcohols exceeds 400 mg/L, they negatively contribute to the wine bouquet [33]. The correlation between the concentration of amino acids and the amount of higher alcohols produced follows a pattern whereby, at levels of yeast assimilable nitrogen (YAN) below 200 mg/L, the production of higher alcohols increases along with the concentrations of YAN, while, above 200 mg/L, the relationship becomes reversible [34,35]. Nevertheless, the concentrations that are reached are strongly related to the strain used [36].

Esters are the most desirable group of compounds contributing fruity and floral aromas to the wine bouquet [36]. *S. cerevisiae* synthesizes two major groups of esters during fermentation, namely the acetate esters of higher alcohols and the ethyl esters of medium-chain fatty acids. Such esters include ethyl acetate, isoamyl acetate, isobutyl acetate and phenylethyl acetate, ethyl hexanoate, ethyl octanoate, and ethyl decanoate [30]. The production of esters during alcoholic fermentation involves the modulation and activation of several genes and their proteic products. In *S. cerevisiae*, six genes and their expression products have been identified to be involved in the ester biosynthesis, as well as hydrolysis ester pathways. Specifically, these genes are *ATF1*, *Lg-ATF1*, * ATF2*, * EEB1*, * EHT1*, and *IAH1* [39,40]. *ATF1*, *Lg-ATF1*, and *ATF2* genes encode for alcohol acetyltransferase (AATases) proteins, which are directly involved in the volatile acetyl ester biosynthesis. *EEB1* and *EHT1* are both involved in the synthesis and hydrolysis pathways of medium-chain fatty acids ethyl esters [39,41]. These genes encode, respectively, for a medium-chain fatty acid ethyl ester synthase/esterase I (*EEB1*) and medium-chain fatty acid ethyl ester synthase/esterase II (*EHT1*). The first enzyme participates in the ethanol acyltransferase and ethyl hydrolase mechanism, the second in the ethanol hexanol-transferase mechanism [39,41]. The *IAH1* gene encodes for esterase enzyme, which contributes to the hydrolysis of acetate esters. Although the precise physiological function of acetyl and ethyl esters is still unknown, the intracellular turnover of these compounds is strictly regulated. Ester production could be a part of a detoxification process for the removal of excess metabolites produced from sugar fermentations [33]. The odors corresponding to the different esters are reported in Table 1. The main ester in wine, ethyl acetate, is desirable at concentrations below 150 mg/L; otherwise, it confers a spoilage character to wine [30]. Taking into account that the concentration of most esters is low and, therefore, very close to the detection limit of human smell in the nose, minimal variations in their concentration can be of great importance for the quality of the final fermentation product (wine) [40,42].

#### 1.2.2. Aldehydes

The major aldehyde synthesized by *S. cerevisiae* during wine fermentation is acetaldehyde, constituting over 90% of the total aldehyde content of wine [41,43]. Acetaldehyde is the last precursor in the anaerobic pathway before ethanol. Acetaldehyde, when present in low concentrations, confers a pleasant fruity aroma, but, when in excess, it produces green and grassy off-flavors [43]. The concentrations of acetaldehyde vary significantly between different types of wine, ranging approximately from 30–80 mg/L in white/red wines to 300 mg/L for sherries [43]. In addition, this compound exerts a direct effect on the wine aromatic profile and an indirect effect due to its high reactivity with other compounds [41,43]. Of special interest is acetaldehyde’s binding activity with SO_2_, the basic antimicrobial agent, which forms a complex compound, consequently limiting the protection to the produced wine [43]. It is noteworthy that different strains of *S. cerevisiae* produce considerably different amounts of acetaldehyde, underlining the fundamental importance of choosing a suitable strain according to the type of wine produced [43,44,45].

In addition, other precursors that do not possess odoriferous characteristics are involved in the development of other aroma substances (e.g., monoterpenes, diols, or terpenes, polyols, fatty acids, carotenoids, glycosylated precursors of aroma, and volatile phenols) [46].

The variability in the formation of volatile compounds during grape must fermentation depends on several factors, including, in particular, the inoculated yeasts, which, transforming the precursors initially present in the must, determine the final organoleptic characteristics of the wine [47]. Recently, a study has reported that wines obtained by different research units, using the same *S. cerevisiae* strains, were generally characterized by different qualitatively volatile molecule profiles in relation to the must employed but also in the function of the inoculated strain [48].

The high strain variability in *S. cerevisiae* is found also in the example reported in Figure 3, which shows the results of experimental wines obtained from the same must fermented by thirty different indigenous strains at laboratory scale. In this experimental trial, principal component analysis (PCA) was carried out on the main secondary compounds of alcoholic fermentation, in particular esters and acetates, alcohols and benzenoids, and other compounds influencing organoleptic characteristics of the wine. The first two components account for about 98% of the total variance. The first principal component (PC1) explained 90.12% of the data variability and was correlated mainly with compounds such as 4-OH-ethylbutyrate, N-(3-methylbutyl) acetamide, ethyl-3-phenyl-2-OH-propionate, N-2-phenylethylacetamide, and citronellol, whereas esters and acetates contribute more strongly to the second principal component (PC2) [47]. The PCA plot revealed that the wines obtained from the same must fermented by the thirty different indigenous strains differed significantly in terms of their aromatic profile as they were located in different quadrants. This result confirms the effective impact of yeast strains on the aroma properties of the wines produced, starting from the same grape must [48].

In this perspective, the selection of new *S. cerevisiae* strains, able to impart specific sensorial and aromatic features, could be a useful tool to increase the gamma of products present on the market. Moreover, in addition to the ability to produce specific volatile profile fingerprinting [49], some other technological features, such as the ability to release mannoprotein, could be a criterion of selection of *S. cerevisiae* strains in order to increase the molecule volatile profiles. In fact, mannoproteins, constituted by mannose (85–90%) and proteins (10–15%), in addition to some positive effects on the technological properties of wines (such as improving the wine colloidal stability, reducing the tannin self-aggregation preventing the reduction of the wine acidity, reducing the protein precipitation promoting the color stabilization), can also affect the release of volatile compounds, affecting the final perception of the wine [50]. In winemaking, the addition of commercial products rich in mannoproteins can be performed, but the challenge to find mannoprotein-producing yeast can represent a great advantage also from an economical point of view. Generally, the use of mannoproteins and fine lees increased the levels of fruity esters, such as ethyl hexanoate, methyl, and ethyl hexadecanoate, probably due to the esterification of fatty acids released by yeasts during fermentation or autolysis [51].

## 2. Production of Aromatic Compounds in Relation to the Main Compositional and Process Variables

The production of aromatic compounds that develop during fermentation processes is strongly influenced both by intrinsic parameters of the raw material, such as the presence of varietal aroma compounds, the qualitative and quantitative composition of the carbonaceous (sugars) and nitrogen (proteins, peptides) fraction, amino acids and inorganic nitrogen, both from the metabolic characteristics of the yeast strain used and from the fermentation parameters (fermentation temperature used, level of micro-oxygenations, etc.), as well as from the refining and aging practices. At the industrial level, the tendency is to optimize the fermentation process in order to enhance the intrinsic characteristics of the raw material, choosing the most appropriate winemaking conditions and fermentation agent, as well as in relation to the type of product to produce [18].

### 2.1. Effect of the Carbon Source on Fermentation Processes and Aroma Compounds

In addition to carbon, nitrogen is of fundamental importance in the yeast cell for growth, cell division, and the production of secondary metabolites with a high aromatic impact. Yeasts are able to assimilate nitrogen both in amino acid form and in ammonia form. High concentrations of intracellular and extracellular nitrogen induce, in *S. cerevisiae* by the metabolic regulation protein complex TOR (target of rapamycin), a state of catabolic inhibition by nitrogen, known as nitrogen catabolite repression (NCR). Nitrogen inhibition determines both the inactivation of the genes coding for permeases and enzymes responsible for the absorption of nitrogen sources present in the extracellular space and the modulation of energy catabolism, stimulating glycolysis and tricarboxylic acids cycle [52]. When nitrogen concentrations are lowered, the catabolic inhibition of nitrogen is canceled by forcing the yeast to use alternative nitrogen sources. In cellar practices, to meet the nitrogen requirement, readily available nitrogen is usually used, in the form of ammonium or amino acids, to reach a concentration of at least 150 mg/L in the medium with a progressive increase in relation to the sugar concentration [53]. Depending on the raw material (fruit or cereals in fermentation processes other than winemaking) and on the *S. cerevisiae* strain used in the fermentation process, the amino acids will be consumed by the medium in an order of preference. Typically, glutamic acid, aspartic acid, asparagine, glutamine, serine, threonine, and arginine are elective nitrogenous sources for *S. cerevisiae* [54]. The production of secondary metabolites with effects on the sensory profile, such as higher alcohols, acetic esters, and sulfur metabolites, is closely linked to nitrogen metabolism. Higher alcohols, in fact, can be formed through the Ehrlich metabolic pathway in which the formation of these compounds directly involves the amino acids leucine, valine, isoleucine, phenylalanine, tyrosine, and tryptophan [22]. Therefore, the nitrogen concentration and the stoichiometric balance of the amino acids present in the matrix directly influence the formation of higher alcohols and acetic esters. In addition to being flavored compounds themselves, higher alcohols are, in fact, also precursors of acetic esters [53,55]. Although many studies have been carried out in model systems, the gene products and metabolic pathways involved in the formation of aroma compounds by *S. cerevisiae* during fermentation processes are known, whereas the exact relationship between the balance of carbon sources and the dynamics that occur in the formation of the compounds has not yet been fully clarified [42].

### 2.2. Effect of the Fermentation Temperature on Fermentation Processes and Aroma Compounds

The fermentation temperature is one of the technological process parameters capable of influencing the metabolism of the yeast and the aromas deriving from it. White wines are generally produced at lower fermentation temperatures than red wines to preserve the fruity and fresh notes typical of these wines. In fact, it has been shown that higher concentrations of esters are obtained for fermentation temperatures between 15 and 18 °C. This is due to the greater stability of the esters at these temperatures, the reduced loss due to evaporation, and the different metabolism of the yeast due to the modulation of membrane fatty acids. However, although the fermentation temperature significantly affects the growth of yeast and its central metabolism, the impact of this process parameter and the biosynthetic pathways associated with the increase in aroma are not fully known. In fact, although it is proven that high concentrations of esters are obtained at low fermentation temperatures, the effects on the production of higher alcohols are still poorly elucidated and more controversial. For example, some authors have shown that only the concentration of 2-phenylethanol, a higher alcohol attributable to the rose aroma, increases as the fermentation temperature increases. Other authors have shown, for all higher alcohols, a tendency to increase as the fermentation temperature increases. Some authors, using synthetic musts and fermentation temperatures of 15 and 28 °C, which, respectively, simulate the vinification temperatures of white and red wines, have analyzed the profiles in volatile molecules of the commercial yeast *S. cerevisiae* EC1118, considered an aromatic reference strain [56]. The results in the synthetic system, where other compositional and process variables have been excluded, have highlighted how the generation of some compounds (for example, hexanol) is related to the presence of precursors with six carbon atoms and the fermentation temperature. The greatest differences detected were identified for ethyl esters, present at higher concentrations in synthetic wines obtained at lower temperatures [56]. On the other hand, four flavoring compounds having a positive contribution on the generation of a good flavor profile, such as 2-methyl-acetate and 2-ethyl-methyl-butanoate (fruity aroma of banana and pineapple, respectively) and 2-phenylethanol and 2-phenyl-acetate (fruity aroma in general), were found at higher concentrations in synthetic wines obtained at 28 °C rather than at 15 °C [56]. Therefore, synthetic wines at 15 °C were characterized by aromas of freshness linked to the production of ethyl esters. It is also known that the adoption of high fermentation temperatures can lead to a greater loss by evaporation of aromatic compounds compared to low temperatures. In fact, this evaporation loss affects the concentration of low-boiling compounds, such as short-chain ethyl esters, to a greater extent than those with higher boiling temperatures, such as medium-chain ethyl esters. In any case, the literature data have highlighted that the generation of different aromatic profiles is more influenced by the central metabolism of the yeast and by the modulation of specific genes with respect to the vinification temperatures adopted, especially in the second half of the fermentation process. Significant linear correlations were found between the concentrations of acetate esters and the respective precursor alcohols, regardless of the fermentation temperatures adopted. This indicates that the degree of synthesis of the acetates depends on the availability of the alcohols. In general, the literature data have highlighted how the temperature, as a technological process parameter, is a very important process variable capable of contributing to the generation of peculiar aromatic profiles in relation to the raw material available, the yeast used, and the specific market demands. For example, Parpinello et al. [57] investigated the potential of the cryotolerant yeast *Saccharomyces eubayanus* to ferment Chardonnay must at different temperatures (10 °C, 12 °C, 16 °C, and 26 °C) over two vintages (2013 and 2014), highlighting also the volatile molecule profile in relation to the used temperature. The effect of added nitrogen was also evaluated. *S. eubayanus* showed its best fermentation performance at low temperatures (10 °C and 12 °C), with optimal kinetic parameters and high sugar consumption. The performance of *S. eubayanus* (EU) was compared with that of a commercial strain of *S. cereviaise* (CE). In particular, in the EU wine, 2-phenethyl alcohol (rose aroma) reached the highest OAV (odor activity value) compared to other aromatics, regardless of the temperature. Among esters, the ethyl nonanoate (nut, rose) was detected in the EU wine only, with a higher OAV in the wine fermented at 16 °C. The ethyl decanoate (fruit, grape) displayed an opposite trend in the EU and CE wines at both temperatures. Ethyl myristate (floral) was detected only in the EU wine produced at 10 °C, whereas methyl myristate (floral, orris) was produced at the highest concentrations by both yeasts at 16 °C. As expected from manufacturer information, CE wines reached high concentrations of ethyl hexanoate (apple, pineapple) and octanoic acid (fruit), with OAVs in the interval 6237–10471. Isoamyl octanoate (fruit) was found only in EU and CE wines at 10 °C. At 10 °C, the EU strain produced less ethyl acetate (which has negative aroma characteristics) than the CE strain. Interestingly, the EU wines were also characterized by a higher concentration of phenethyl acetate (rose) than CE wines at both temperatures; this aromatic compound is considered a marker in fermentation performed with *S. eubayanus* [58]. Isoamyl acetate (banana, fruit) was the dominant compound for CE wines at both temperatures; this result was expected since the strain has been genetically selected to prioritize ester production (reaching the highest level at 16 °C). For EU wines, the concentrations of this compound were comparable to CE wines at 10 °C (regardless of the nitrogen supplementation) and at 16 °C without nitrogen supplementation, supporting previous findings that high isoamyl acetate production occurs at low temperatures [10].

### 2.3. Effect of Non-Thermal Technologies on the Production of Wine Aroma Compounds

During the last decade, the application of non-thermal technologies to produce, age, and preserve wine has become an area of great interest. In particular, several authors have underlined the potential of high-pressure homogenization (HPH), pulsed electric fields (PEF), high-pressure processing (HPP), ultrasound (US), low electric current (LEC), pulsed light (PL), ultraviolet irradiation, and filtration applied to the must or wine as stabilization methods [59,60].

In particular, PEF and HPH are considered as promising technologies for the beverage industry as they are continuous processes, requiring only microseconds or milliseconds, respectively, of processing time for the inactivation of undesirable microorganisms in wines, with commercial-scale and higher throughput production potential. However, some authors have underlined the role of some non-thermal technologies, applied at sublethal level to bacteria or yeast cells used as culture starters, in order to modify their metabolism and, consequently, the production of aroma compounds, as reported by Figure 4.

For example, some authors have hypothesized how HPH, when applied to bacteria or yeast cells at levels ranging between 50 and 100 MPa, induces changes in gene expression that regulate specific properties (such as autolysis or production of aroma compounds) without affecting the cell viability or the fermentation kinetics [61,62,63,64]. In addition, to modify their metabolism, the application of HPH at a sub-lethal level was suggested also to modify some functional properties of lactic acid bacteria to be used in the dairy sector, with such exposure of protein hydrophobic regions in the cell wall resulting in different autoaggregation and adhesion capacities [65]. In the oenological field, Patrignani et al. [62] showed that an HPH treatment applied at 90 MPa to different yeasts (*Saccharomyces bayanus* L951 and *S. cerevisiae* ML692 and commercial strains *S. bayanus* Lalvin CH14, *S. bayanus* IOC 18-2007, *S. bayanus* Lalvin EC1118, and *S. bayanus* IT 1818), prior to use or prior to their use in the preparation of tirage solutions for sparkling wines, induced an acceleration of yeast autolysis with a consequent release of mannoproteins and different patterns in volatile profiles. In fact, the SPME-GC-MS and electronic nose analyses indicated significant changes in volatile patterns due to HPH treatment. In fact, the sparkling wines produced with HPH-treated cells, with the exception of those by strain L951, were significantly different on the basis of PLS analysis from the control wines. The wines obtained from treated cells were characterized by high acid contents and, particularly, short-chain fatty acids, associated with low levels of sulfur compounds. The wines obtained from HPH-treated strains had significantly modified ester profiles, with lower concentrations of those compounds with medium- and long-chain fatty acids with respect to the control wines. Moreover, the autolytic patterns of the HPH-treated cells can contribute to the different volatile profile recorded. The volatile profile of sparkling wine is reported to be dependent on the kinetics of volatile compound retention and release by yeast less during the aging process [66]. The ability of yeast lees to interact with volatile compounds depends on lees composition, which, in turn, depends on yeast species, strain, and growing conditions [67]. However, the HPH treatment is reported to modify macromolecules, such as polysaccharides and proteins, with consequent modification of the susceptibility to enzymatic attack and to interact with other molecules [68]. According to Gallardo-Chancon et al. [69], the lee sorption of volatiles was proportional to their hydrophobicity, and their retention by the lees surface changed during aging. Both the glucan layer and the proteic component resulting from the hydrolysis of mannoproteins, as well as the inner plasma membrane, participate in sorption of hydrophobic molecules due to the increased porosity of the yeast cell wall during autolysis [70]. While most polar aromas were released from the lees surface at the earliest stages of aging, highly hydrophobic compounds and esters were progressively retained until ~18 months and subsequently desorbed into wine [69].

The accelerated autolysis and the increased exposure of the hydrophobic region of protein due to HPH treatment [71] can account for the lower content of hydrophobic compounds in the sparkling wine obtained from HPH-treated cells. However, changes in the activity and specificity of several enzymatic systems and the cell response mechanisms due to HPH treatment can contribute to changes in volatile compound profiles. The enhancement of enzyme activities and the change in their specificity in HPH-treated cells have been previously reported for yeast and lactic acid bacteria [63]. The application of sublethal stress is reported to modify gene expression, and, consequently, microbial metabolism, throughout the production of volatile signaling compounds, with a great impact on food sensorial features. Gottardi et al. [64] proved that the application of an HPH sub-lethal treatment at 100 MPa on the wine starter *S. cerevisiae* ALEAFERM AROM grown in synthetic must induced a gene expression response of HPH-treated cells with a massive rearrangement of gene expression due to the identification of 1220 differentially expressed genes (DEGs). Most of the genes related with energetic metabolic pathways and ribosome structure were down-regulated, while genes involved in ribosome maturation, transcription, DNA repair, response to stimuli, and stress were up-regulated. Moreover, the samples produced by HPH-treated cells, compared to the control samples (cell not HPH-treated), after 48 h from the inoculation, were characterized by higher production of 2-phenylethanol, together with a significant increase in benzaldehyde (able to confer almond notes), ethanol, isoamyl alcohol (alcoholic, winey, fruity notes), acetic acid, and ethyl octanoate (fruity, fat aroma). Moreover, Comuzzo et al. [72] showed an increase in alcohols, particularly 2-phenylethanol, and esters, mainly ethyl octanoate, in *S. bayanus* powder after HPH treatment, confirming the changes in the volatile molecule patterns. Moreover, according to the authors, the presence of higher amounts of alcohols and ethyl esters also represented evidence regarding the ability of HPH to induce autolysis in wine yeasts. These findings suggest that HPH induces or promotes an autolytic-like behavior that can be exploitable in winemaking also for the differentiation of the gamma of products. Some authors also reported, the use of some non-thermal technologies, such as high-pressure homogenization (HPH), for the treatment of yeast cells also in scalar fermentation.

In the Figure 5a,b is reported the projection of wine samples and volatile compounds of wine samples obtained by scalar fermentation of a wild strain of *Candida zemplinina* (CZ) and a commercial strain of *S. cerevisiae* (SC) after 21 days of fermentation and in relation to the initial HPH treatment.

## 3. Aromatic Compounds and Sensory Profiles of Wines in Relation to the Interactions between *Saccharomyces* and Non-*Saccharomyces* Yeasts

The species *S. cerevisiae* certainly represents the main yeast of alcoholic fermentation in relation to its technological performances, such as the ability to ferment the sugars of the grape must quickly and the ability to survive at low pH values and high alcohol levels. Due to these characteristics, it is the prevalent yeast present at the end of the vinification process. On the other hand, the presence of *S. cerevisiae* strains on grapes is very low in comparison to non-*Saccharomyces* yeasts, which can start the fermentation process but, due to their low alcohol tolerance, are usually replaced by *S. cerevisiae* [73,74]. For many years, non-*Saccharomyces* yeasts have been associated with a negative role in winemaking because many strains were linked to phenomena of alteration, mainly due to their ability to increase the volatile acidity of wines. Nowadays, it is widely recognized that non-*Saccharomyces* yeasts contribute to the fermentative process and improve the aroma profile of the final wine despite their low fermentative power. In the past, they were considered as spoilage agents for the high production of undesirable metabolites, such as acetic acid, ethyl acetate, and off-flavors, resulting in anomalous analytical and sensorial profiles of the wine [75]. However, in the last few decades, non-*Saccharomyces* yeasts were positively reevaluated in winemaking due to their ability to produce aromatic compounds during the first stages of fermentation that contribute to enhancing the wine complexity. They are considered as a potential biotechnological tool, in mixed fermentations with *S. cerevisiae*, for their metabolic and enzymatic activities affecting the formation of the final bouquet [76,77]. In fact, in the early stages of alcoholic fermentation, non-*Saccharomyces* yeasts hydrolyze the glycosidic bonds of the odorless, non-volatile glycoside, determining the release of the aromatic component as terpenols, norisoprenoids, and alcohol benzoics due to the β-glucosidase activity. The use of non-*Saccharomyces* yeasts could satisfy both the growing need of winemakers to innovate and diversify wine production in order to better meet the request of the globalized market and the interest of the consumers oriented towards increasingly aromatic products but with lower alcohol levels than in the past [78]. Nowadays, the increase in alcohol levels in wine is one of the main problems affecting the winemaking sector due to high sugar concentrations in grape musts arising from global warming [79]. Because of these advantages, some non-*Saccharomyces* yeasts belonging to *Torulaspora delbrueckii*, *Lachancea thermotolerans*, * Metschnikowia pulcherrima*, * Schizosaccharomyces pombe*, * Starmerella bacillaris* (synonym *Candida zemplinina*), and *Pichia kluyveri* are marked by the important manufacturers [80,81] and used at the industrial level to increase the wine aroma profile [20]. In both inoculated and spontaneous fermentations, the role of non-*Saccharomyces* yeasts is fundamental in the early stages of alcoholic fermentation, before *S. cerevisiae* becomes dominant. In fact, yeasts contain genes that code for specific enzymes responsible for maintaining their survival in the ecosystem in which they are present. Some of these enzymes catalyze reactions involved in the transformation of sugars into ethanol; others are involved in the formation of the primary or secondary aromas described in the previous paragraphs. Due to the high biodiversity of the indigenous non-*Saccharomyces* population in the initial grape must, the range of extracellular enzymes produced during a spontaneous fermentation is much wider than that obtained by the inoculation of a *S. cerevisiae* starter. The most modern genetic sequencing techniques have shown that the non-*Saccharomyces* population is able to produce a greater variety of extracellular enzymes than *S. cerevisiae*. Consequently, isolation and characterization of these yeasts have become areas of research of great interest in the oenological field for the development of starters and co-starters, with the aim of innovating and improving wine production, even the most traditional [82].

The characteristics of the most employed non-*Saccharomyces* yeasts in terms of flavor production are reported below and summarized in Table 2.

***Torulaspora delbrueckii*** is one of the most studied non-*Saccharomyces* species in winemaking, being a yeast with interesting metabolic properties, and it is already marked as active dry yeast [81]. It contributes to the overall quality of the wine because it typically produces low concentrations of acetic acid [83], one of the main parameters of oenological relevance. It has been claimed that *T. delbrueckii* allows to increase the glycerol content from 0.2 to 0.9 g/L [84,85,86] and to produce high amounts of diacetyl, ethyl lactate, and ethyl acetate compared to the fermentation conducted by *S. cerevisiae* alone [87]. Moreover, in wines obtained by mixed fermentation (*T. delbrueckii* and *S. cerevisiae*), a high content of acetate esters, medium-chain fatty acids, thiols, α-terpineol, and linalool, which is derived from monoterpene alcohols, were found. *T. delbrueckii* can produce higher levels of fruity esters and terpenes and lower amounts of higher alcohols when used in sequential fermentations with *S. cerevisiae*, respecting the varietal character of the grape [88,89,90]. Some authors reported that *T. delbrueckii* is characterized by high levels of extracellular enzymes, such as β-glucosidase, with an impact on the wine sensory profile, by modulating the levels of norisoprenoids, terpenols, and lactones in consequence of the hydrolysis from their respective precursors [91,92]. The use of *T. delbrueckii* strains can decrease the final ethanol concentration in wines by about 1% (*v/v*) [91,93] more than traditional fermentations [94]. Other metabolites produced include succinic acid and a notable mannoprotein and polysaccharide release ability in wine, which increases the quality of wine with an influence on mouthfeel properties [95]. The management of *T. delbrueckii* is easy compared to other non-*Saccharomyces* yeasts for its relatively high fermentative power and the ability to tolerate ethanol concentration up to 9–10% (*v/v*). Due to the high ethanol resistance, this species influences the characteristics of final wine during almost all the alcoholic fermentation, although *S. cerevisiae* is necessary to complete the fermentative process.

***Lachancea thermotolerans*** is actually marked by important manufacturers and used to acidify grape juices [96] because it is the unique yeast that produces lactic acid during its fermentative metabolism [97]. The reduction in pH is a very useful strategy, mainly in the warm viticulture region, and also to influence the color of red wine, increasing the anthocyanins color intensity [93]. *L. thermotolerans* has other important oenological characteristics, such as the ability to reduce the volatile acidity in wine [98] by producing lower concentrations of acetic acid of about 0.24 g/L compared to *S. cerevisiae* [99,100]. Additionally, among the non-*Saccharomyces* species, *L. thermotolerans* shows moderate ethanol resistance (10% (*v/v*)) [99,101]. In fact, it is considered a good fermentative species, but, in order to complete the alcoholic fermentation, it must be combined with a *S. cerevisiae* strain [102]. Some studies reported that mixed fermentations with *L. thermotolerans* and *S. cerevisiae* reduce ethanol concentrations, ranging from 0.2% to 0.4% (*v/v*) [19,93]. Furthermore, it was demonstrated that *L. thermotolerans* enhances floral and tropical fruit aromas during mixed fermentations of grape musts, producing wines with high concentrations of glycerol and 2-phenylethanol [103].

***Metschnikowia pulcherrima*** is actually available on the market and commonly associated with grapes and wine [103]. In recent years, it has demonstrated its influence on the sensory profile of wine [104] due to some oenological characteristics, such as: high β-glucosidase activity, ability to decrease the volatile acidity and to increase the production of compounds influencing the organoleptic quality of wine, such as medium-chain fatty acids, higher alcohols, esters (mainly ethyl octanoate), terpenols [105], and glycerol, when used in mixed fermentations. Strains of this non-*Saccharomyces* species are characterized by an extra-cellular α-arabinofuranosidase, influencing the content of desirable varietal aromas, such as terpenes and volatile thiols. In particular, *M. pulcherrima* releases varietal thiols, such as 4-methyl-4-sulfanylpentan-2-one, in concentrations higher than those produced by *S. cerevisiae* due to the cystathionine-β-lyase activity [93]. Studies have reported that *M. pulcherrima* is also able to improve the levels of methyl butyl-, methyl propyl-, and phenethyl esters [91,103]. In addition, it has also been observed that this species might have an antagonistic effect toward several yeasts, including *S. cerevisiae*, with delays in fermentation. This effect was correlated to the killer character as a consequence of the production of pulcherrimin pigment by *M. pulcherrima* [91]. This species was recently proposed for the reduction of ethanol. It has been used in mixed fermentations, both in sequential and simultaneous inoculation conditions, obtaining a reduction of ethanol by about 1.6% [81,104,106].

***Schizosaccharomyces pombe*** is usually used for the deacidification of musts, mainly from cool areas, such as those from the north of Europe, as it can metabolize malic acid into ethanol and CO_2_, reducing the total wine acidity [91]. It is present as a commercial product as an alternative method for wine deacidification. The wines obtained by mixed fermentation showed increased concentrations of acetaldehyde, propanol, and 2,3-butanediol but lower concentrations of higher alcohols and esters [103]. This characteristic is very interesting for wine in which the preservation of the varietal aroma of grapes is desired more than the fermentative aroma [93]. *Sch. pombe* releases higher amounts of polysaccharides (α-galactomannose and β-glucans) [93,107,108], consequently improving the wine structure. *Sch*. *pombe* can be used as a new tool to assure wine safety because it could be used to reduce the urea content (precursor of ethyl carbamate, a toxic compound in wine) by the urease activity [109]. However, one of the main problems related to the use of *Sch. pombe* is the risk of production of high levels of acetic acid [108]. Different strategies have been tested to reduce this undesirable effect, such as the mixed fermentation with *S. cerevisiae* or *T. delbrueckii* or the use of cells immobilized in alginate. Another undesirable effect of the use of *Sch. pombe* is an increase in the ethanol concentration as the degradation of malic acid produces additional ethanol [91,103].

***Pichia kluyveri*** is studied by wine researchers because it has been shown to release flavor precursors from grape juice, with a potential enhancement in the wine aroma and flavor. Mixed fermentation with *P. kluyveri* has been reported to lead to higher levels of varietal thiols, especially 3-mercaptohexyl acetate (3MHA), 2-phenylethyl acetate, and ethyl octanoate. Moreover, an increase in the total terpene concentration has been reported, improving the grape variety typicity [93,103].

***Hanseniaspora***. The yeasts belonging to this genus show a typical apiculate shape, and, among these, four species are associated with winemaking: *H. guilliermondii*, *H. osmophila*, *H. vinae*, and *H. uvarum*. They compose the common microbiota of grape berries and are found in the highest numbers in grape must [103]. Therefore, they play a significant role at the beginning of the spontaneous fermentation until alcohol levels of about 4–6% are reached. At these levels, most *Hanseniaspora* strains cannot survive due to their low tolerance to ethanol; however, they contribute significantly to the character and quality of the final wine [104]. The *Hanseniaspora* species, in particular *H. guilliermondii*, *H. uvarum*, and *H. vinae*, are good producers of interesting enzymes in winemaking, such as β-glucosidase, β-xylosidase, and protease, in particular for their application at an industrial scale. These enzymatic activities are correlated with the higher production of 2-phenylethyl acetate, often associated with rose, honey, and flowery sensory descriptors [74], acetate esters (such as isoamyl acetate), medium-chain fatty acid, ethyl esters, and terpenes, and a low amount of higher alcohols [93,103], improving the aroma profile of certain styles of wine. Nevertheless, apiculate yeasts may also be associated with the production of undesirable flavor compounds, such as high levels of volatile acidity and sulfur compounds. The wine species of the genus *Hanseniaspora* generally exhibit low fermentative power, but the use in mixed fermentation with *S. cerevisiae* allows the production of some interesting volatile compounds. For example, co-fermentation of must with *H. osmophila* and *S. cerevisiae* has produced wines exhibiting an increase in fruit sensory compared to wines produced with *S. cerevisiae* monoculture, while wines produced with *H. uvarum/S. cerevisiae* showed increased concentrations of higher alcohols, acetate, and medium-chain fatty acids [110].

***Starmerella bacillaris*** (synonym *Candida zemplinina*) positively contributes to the production of secondary metabolites, and, in particular, it is known as a high glycerol producer, with concentrations in wine up to 14 g/L during alcoholic fermentation [111]. These concentrations can improve the mouthfeel sensation and flavor of wine [112]. Another interesting property is its fructophilic character [113], which is the preferential consumption of fructose rather than glucose [94,108], unlike *S. cerevisiae*, which preferably utilizes glucose [103]. *St. bacillaris* persists up to the middle-end phase of the fermentation process due to its ability to tolerate high concentrations of ethanol [114]. In mixed fermentations, if used in co-inoculation with *S. cerevisiae*, it is not able to consume all the sugar in the must, reaching ethanol levels of about 5.8% (*v/v*) [87]. On the contrary, in fermentations by sequential inoculation, this yeast allows to obtain dry wines with reduced ethanol content and low acetic acid production. In addition, some studies report a higher production of glycerol in fermentations by sequential inoculation (15.7 g/L) compared to simultaneous fermentations (5.2 g/L) [87]. Generally, *St. bacillaris*/*S.cerevisiae* wines are characterized by high concentrations of linalool, citronellol, nerolidol, geraniol, and terpenes, with a consequent increase in aroma complexity [73].

***Wickerhamomyces anomalus***, also known as *Pichia anomala*, is another yeast frequently isolated from grapes and must. Some studies have reported that *W. anomalus* produces a very high level of extracellular enzymes, such as β-glucosidases and β-D-xylosidases, a high level of monoterpenes, and a high level of fruity acetate esters, such as 2-phenylethylacetate [115]. It is quite resistant to ethanol, but its use as a fermentative yeast is limited by the very high production of ethyl acetate, which confers an unpleasant solvent-like aroma in wine, even in co-inoculation with *S. cerevisiae* [116].
foods-11-01921-t002_Table 2Table 2Main non-*Saccharomyces* species of oenological interest and their contribution in aromatic compounds in wine.YeastCharacteristics***Torulaspora delbrueckii***Low production of acetic acid [83]Increase in glycerol content [84,86]High production of diacetyl, ethyl lactate, and ethyl acetate [87]***Lachancea thermotolerans***Production of lactic acid during its fermentative metabolism [97]Increase in the anthocyanins color intensity [93]Reduction in volatile acidity [98]Low production of acetic acid [99,100]Moderate ethanol resistance [99,102]***Metschnikowia pulcherrima***High β-glucosidase activity [77]Release of varietal thiols [93]Improvement in some esters [91,103]***Schizosaccharomyces pombe***Deacidification of musts metabolizing malic acid [91]High release of polysaccharides [93,107,108]High production of acetic acid [108]***Pichia kluyveri***Release of flavour precursors from grape juice [103]Increase in terpene concentration [93,103]***Starmerella bacillaris***High production of glycerol [111]Fructophilic character [113]High ethanol tolerance [114]***Hanseniaspora*****spp**.(*H. guilliermondii, H. uvarum, H. vinae*)Production of β-glucosidase, β-xylosidase, protease [93]High production of 2-phenylethyl acetate [74]Low production of higher alcohols [93,103]***Wickerhamomyces anomalus***High production of β-glucosidases, β-D-xylosidases, monoterpenes [112]High production of fruity acetate esters [115]High production of ethyl acetate [116]


Of particular importance for non-*Saccharomyces* yeasts is the β-glucosidase activity, carried out by these yeasts in the early stages of alcoholic fermentation, hydrolyzing the glycosidic bonds of the odorless non-volatile glycoside and determining the release of the aromatic component, which consists of terpenols, norisoprenoids, and benzoic alcohols. Generally, monoterpenic glycosides in grapes are molecules linked to a sugar and, therefore, are non-volatile and odorless, but they can be used as potential indicators of quality. Grapes may naturally contain β-glycosidase, whose activity is, however, minimal due to the fermentation temperature, low pH, and high sugar concentration of grape must. For this reason, in recent years, the selection of wine yeasts with positive glycosidase activity has been extensively applied, with the aim of enhancing the biotechnological potential of the grape microbiota or of inoculated yeasts in order to increase the varietal aroma without resorting to the addition of exogenous enzymes. In this regard, many non-*Saccharomyces* yeasts, albeit producing low levels of ethanol, produce β-glucosidase, whose activity, although influenced by the chemical–physical characteristics of the must and the technological parameters of winemaking, is able to improve the wine sensory profile in a decisive way. Obviously, even the species considered and especially the characteristics of the strain used can significantly affect the persistence and activity of these enzymes in must and wine. Otherwise, this character is poorly expressed between *S. cerevisiae* and *S. bayanus* strains, and few strains of the genus *Saccharomyces* are able to carry out both the primary oenological function (alcoholic fermentation) and the production of enzymes capable of enhancing the varietal aroma. Wide interest is being shown by many researchers and operators in the oenological field on the use of sequential inoculation, whose goal is to imitate spontaneous fermentation by inoculating non-*Saccharomyces* species at the early stage and successively *S. cerevisiae*, with the aim to guarantee the expression of the metabolism by non-*Saccharomyces* at the initial phase of the fermentation process, increasing the overall aroma of the wine. In any case, the selection and choice of the non-*Saccharomyces* starter to be inoculated are crucial for the quality of the final product, also according to the sensory characteristics that are to be conferred to the wine. Furthermore, it should be emphasized that even the inoculum levels between non-*Saccharomyces* and *Saccharomyces* in relation to the winemaking conditions can strongly influence the profile of the final aroma, as reported in Figure 6.

## 4. Conclusions

The literature data have long established that taste is the primary choice and driver factor for wine consumption among other factors, such as emotion, social/peer pressure, and human physiological factors. In turn, the final wine taste can be related to several aspects, among which the aromatic wine complexity depends on countless factors and their complex interactions [117]. One of the most important interactions described in this review is that between *S. cerevisiae* strains and non-*Saccharomyces* ones, or the presence of mannoproteins affecting the release of aromatic components. However, in order to enlarge wine production and its taste, one of the most ambitious challenges of the modern wine industry is to innovate and diversify wine production processes in order to better meet the needs of the globalized market and consumers oriented towards increasingly aromatic products but with low alcohol content compared to the past. However, the sensory appeal of a wine always remains the first driver of consumer choice and, consequently, the agronomic practices adopted in the field for the cultivation of the vine, along with the choice of strains to be used in fermentation, as well as the parameters of the winemaking process and aging of the wine need to be aimed at achieving this goal. According to the findings of this review, one of the most important tools to increase the aroma compounds in wine is the suitable use of microbial biodiversity. In fact, as previously shown in Figure 3, the exploitation of the high strain variability in *S. cerevisiae* species is a useful example to modulate the final wine aroma compounds. Moreover, other important factors are the proper use of non-*Saccharomyces* yeasts in the generation of wine aroma compounds and the enhancement of the biodiversity of the grape microbiota to select strains with specific attributes that, if used wisely, can contribute in a positive way to the improvement and characterization of the aromatic profile of the wine. In particular, this can be due, as underlined before, to the β-glucosidase activity carried out by these non-*Saccharomyces* yeasts in the early stages of alcoholic fermentation, determining the release of the aromatic component, which consists of terpenols, norisoprenoids, and benzoic alcohols. Another interesting aspect underlined in this review is the set-up of protocols based on non-thermal technologies (such as high-pressure homogenization) useful to treat the initial starters, modifying their metabolism and the release of secondary metabolites.

However, to achieve innovation in a mature sector, such as the oenological one, it will be necessary to strengthen the collaboration between the oenological world and scientific research so as to ensure an effective industrial implementation of the results obtained on a laboratory scale.

## Figures and Tables

**Figure 1 foods-11-01921-f001:**
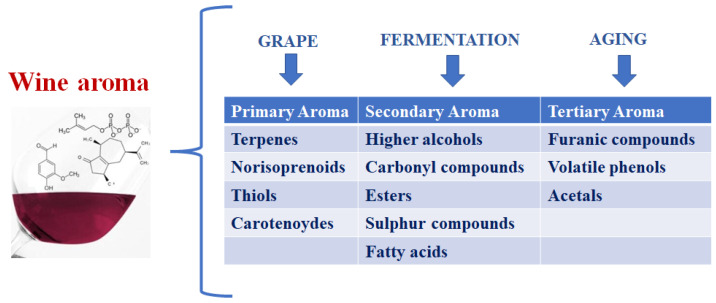
Subdivision of the wine aroma complexity according to its origin.

**Figure 2 foods-11-01921-f002:**
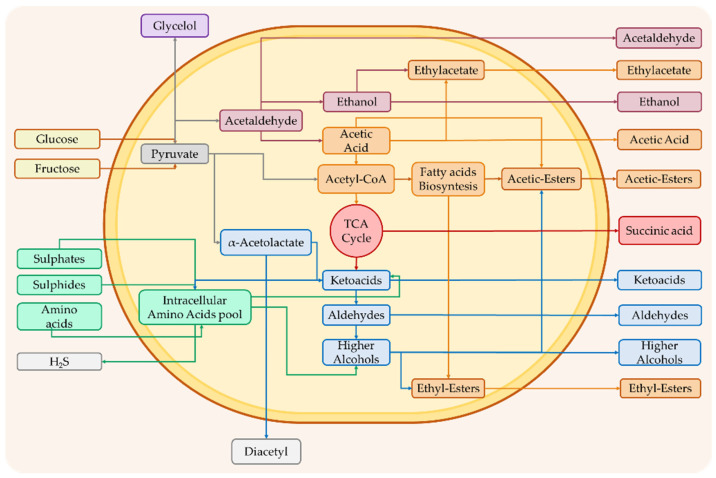
Representation of the main biochemical pathways that lead to the formation of compounds deriving from primary and secondary metabolism in yeast able to affect the aroma of wine.

**Figure 3 foods-11-01921-f003:**
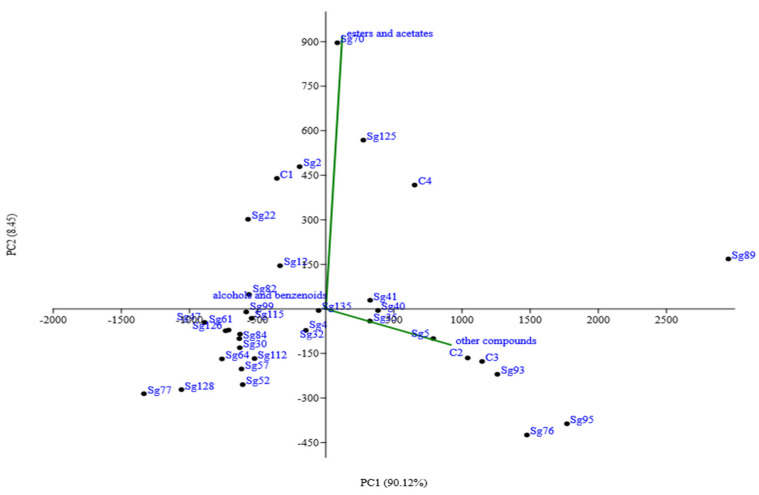
Principal component analysis (PCA) plot based on by-products detected in wines produced at laboratory scale with 30 different indigenous *S. cerevisiae* strains.

**Figure 4 foods-11-01921-f004:**
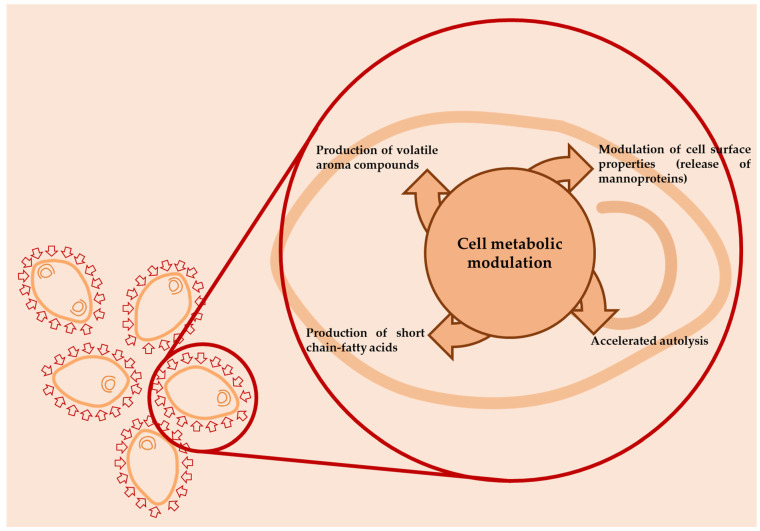
Sublethal HPH process applied to yeast cell to modify cell metabolism, autolysis, and production of volatile aroma compounds.

**Figure 5 foods-11-01921-f005:**
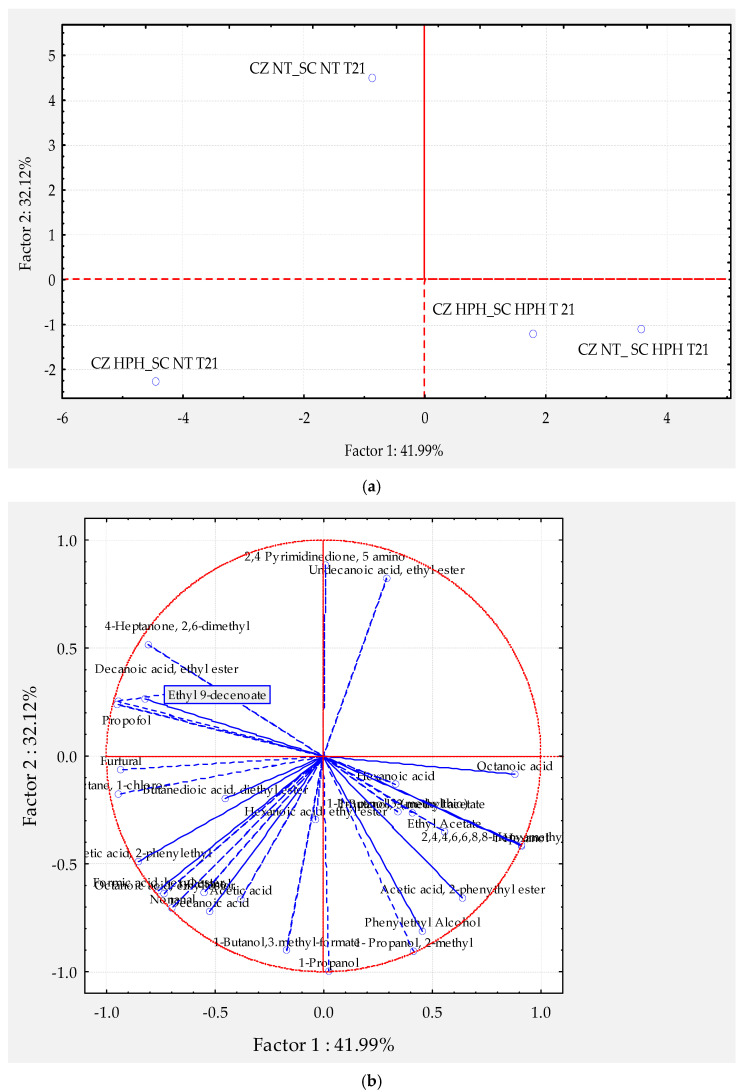
(**a**) Principal component analysis loading plot (PCA) of the PC1 and PC2 of wine samples volatile aroma profiles obtained by scalar fermentation of a wild strain of *Candida zemplinina* (CZ) and a commercial strain of *S. cerevisiae* (SC) after 21 days of fermentation and in relation to the initial HPH treatment; (**b**) Principal component analysis factor coordinates (volatile compounds) mapped in the space described from PC1 and PC2, accounting respectively, the 41.99% and 32.12%, variance among samples observed.

**Figure 6 foods-11-01921-f006:**
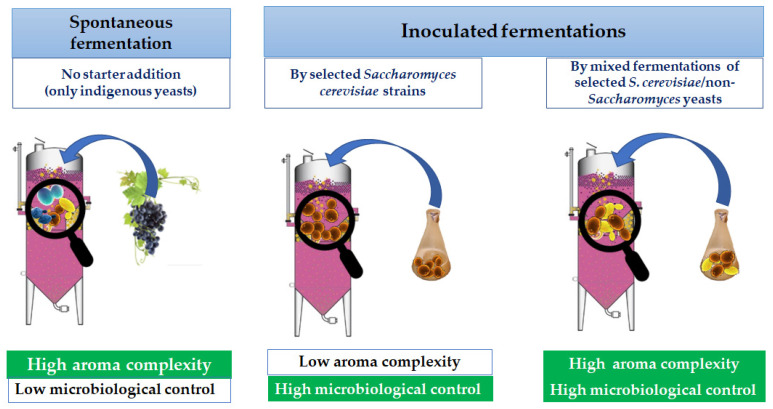
Effects of the presence of non-*Saccharomyces* and *Saccharomyces* yeasts on wine aroma profile.

**Table 1 foods-11-01921-t001:** Concentrations and odors corresponding to the different aroma compounds in wine.

Compound	Concentration in Wine (mg/L) [37]	Aroma Threshold (mg/L)								Aroma Descriptors [37]
		Water		Ethanol (10% *v/v*)		Wine		Synth. Wine		
Ethyl acetate	22.5–63.5	0.005–5	[38]	7.5	[37]					Fruit, solvent
Isoamyl acetate	0.1–3.4	0.002	[38]	0.03	[37]					Banana, pear
2-phenylethyl acetate	0–18.5			0.25	[37]					Floral, fruity, rose
Isobutyl acetate	0.01–1.6	0.066	[38]							Banana, fruity
Exiyl acetate	0–4.8					0.7	[37]			Sweet, perfume
Ethyl butanoate	0.01–1.8	0.001	[38]	0.02	[37]					Floral, fruity
Ethyl esanoate	0.03–3.4			0.05	[37]					Green apple
Ethyl octanoate	0.05–3.8			0.02	[37]					Soap
Ethyl decanoate	0–2.1							0.2	[37]	Floral, soap
Propanol	9–6.8	9	[38]			500	[37]			Pungent, astringent
Butanol	0.5–8.5	0.5	[38]	150	[37]					Alcoholic
Isobutanol	9–174			40	[37]					Alcoholic
Isoamyl alcohol	6–490	0.25–0.3	[38]	30	[37]					Astringent, solvent
Hexanol	0.3–12					4	[37]			Herbaceous
2- Phenylethyl alcohol	4–197			10	[37]					Floral, rose
Acetic acid	100–1150			280	[37]					Acidic, vinegar
Acetaldehyde	10–75	0.015–0.12	[38]			100	[37]			Unripe walnut, bruised fruit, sherry
Diacetyl	<5	0.0023–0.0065				0.2–2.8	[37]			Buttery
Glycerol	5–14					5.2 g/L	[37]			Odorless (slight sweet taste)
Linalool	0.002–0.01	0.006	[38]					0.025	[37]	Rose
Geraniol	0.001–0.044	0.04–0.075	[38]	30	[37]					Rose
Citronellol	0.015–0.042	0.04–0.086	[38]	100	[37]					Citronella
2-Acetyl-1-pyrroline (ACPY)	traces	0.0001	[37]							Mouse urine
2-Acetyltetrahydropyridine (ACPTY)	0.005–0.1	0.0016	[37]							Mouse urine
4-Ethylphenol	0.012–6.5			0.14	[37]	0.6	[37]			Medicinal, stable
4-Ethylguaiacol	0.001–0.44	0.04	[38]	0.033	[37]	0.11	[37]			Phenolic, sweet
4-Vinylphenol	0.04–0.45	0.02	[37]							Drug
4-Vinylguaiacol	0.001–0.71	10	[37]							Cloves, phenolic

## Data Availability

Not applicable.

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
