# Peer review of "Role of Yeasts on the Sensory Component of Wines"

_foods, 2022, doi:10.3390/foods11131921_

Round 1

Reviewer 1 Report

The authors report the influence of yeasts on aromatic composition of wine,  highlighting the relevance of microbial population in the aromaric complexity of wine. They also report the effect of oenological practices and winemaking conditions related to yeasts development and metabolism in the final sensory profile of wines. However, the manuscript needs improvement.

General comments:

-         Abstract, the last phrase should be modified, it seems like S. cerevisiae strains are not of oenological interest.

-         Sections are not numerated correctly, section 2.3. should be change to 2.2., section 2. Effect of non-thermal technologies should be included in section 2 as 2.3,  section where interaction between yeasts should be section 3 and conclusions section 4.

-         Grammar and spelling should be revised, for example, capital letters in lines 490 and 515, line 650,  line 737, line 802 …

Major issues:

-         Some paragraphs should be rewritten or reorganized as the same concept is repeated along them (lines 55 to 63, lines 200 to 210, lines 597 to 609).

-         Remove repeated comments. Lines 226 to 228, previously in lines 208 to 210, lines 232 to 234 and 381 to 384, previously in page 4.

-         In line 593 the authors wrote that S. cerevisiae is the ONLY YEAST present at the end of the vinification process and they are not found in grapes. These statements are not true and should be expressed correctly, other yeasts species can be found at the end of fermentation and the presence of S. cerevisiae in grapes is very low in comparition to non-Saccharomyces yeasts.

-         Remove lines 832-839 and figure 6, it does not make sense, the paragraph and the figure does not contribute to any relevant information in this review.

-         Conclusion part as presented is too general, interesting  key point or key factors should be stated out.

Author Response

Dear Editor

we would like to thank the Referee 1 for his/her careful revision which permitted us to improve the review.

All the suggestions were accepted and the text modified in red according to.

Best Regards

Francesca Patrignani

Reviewer 2 Report

Introduction needs to be further subheaded for the class of compound family. A summarised table 

Table 1. How did the author retrieve the aroma descriptors? Ref needed. Would also be good to include the odor threshold too.

Subheader number needs to be corrected

Conclusion - the author claims that the aromatic complexity is dependent on the factors and interactions. However, this might not be the case with sensory complexity. If the authors were to claim this then there should be a brief discussion on what complexity is in the Introduction section. Similarly to sensory pleasantness that is only mentioned in the conclusion section.

Perhaps a short section on consumer drivers/barrier of wine would assist this conclusion claim.

Author Response

we would lke to thank the Referee 2 for his/her careful revision which permitted us to improve the review. All the suggestions were accepted and the changes are in red in the text.

Thanking you for your support

Best regards

Francesca Patrignani

Round 2

Reviewer 1 Report

The manuscript has been improved. I suggest it to be accepted for publication.

Author Response

we would like to thank the Referee 1

Reviewer 2 Report

Please check if the figures that from a referenced paper need copyright.

The authors haven't elaborated what is wine complexity, a quick search in the manuscript only shows 6 hits - 2 being in reference, 1 in abstract, 2 in the body, and 1 being in conclusion. If the author conclude that the wine can be complex, then the term complexity needs to be elaborated.

Author Response

We would like to thank the Referee to give us the chance to increase the quality of this review.

The concept of wine complexity was introduced in the beginning of the review
